# Combinatorial interactions between viral proteins expand the potential functional landscape of the tomato yellow leaf curl virus proteome

Liping Wang[1,2☉], Huang Tan[1,2,3☉], Laura Medina-Puche[1,3], Mengshi Wu[1,2], Borja Garnelo Gomez[1], Man Gao[1,2], Chaonan Shi[1,3], Tamara Jimenez-Gongora[1,2], Pengfei Fan[1,2], Xue Ding[1,2], Dan Zhang[1,2], Yi Ding[1,2], Tábata Rosas-Díaz[1], Yujing Liu[1], Emmanuel Aguilar[1,4], Xing Fu[1], Rosa Lozano-Durán[1,3]*

1 Shanghai Center for Plant Stress Biology, Center for Excellence in Molecular Plant Sciences, Chinese Academy of Sciences, Shanghai, China, 2 University of the Chinese Academy of Sciences, Beijing, China, 3 Department of Plant Biochemistry, Center for Plant Molecular Biology (ZMBP), Eberhard Karls University, Tübingen, Germany, 4 Instituto de Hortofruticultura Subtropical y Mediterránea "La Mayora" (IHSM-UMA-CSIC), Area de Genética, Facultad de Ciencias, Universidad de Málaga, Campus de Teatinos s/n, Málaga, Spain

☉ These authors contributed equally to this work.
* rosa.lozano-duran@zmbp.uni-tuebingen.de

**Data Availability Statement:** All relevant data are within the manuscript and its Supporting Information files.

## Abstract

Viruses manipulate the cells they infect in order to replicate and spread. Due to strict size restrictions, viral genomes have reduced genetic space; how the action of the limited number of viral proteins results in the cell reprogramming observed during the infection is a long-standing question. Here, we explore the hypothesis that combinatorial interactions may expand the functional landscape of the viral proteome. We show that the proteins encoded by a plant-infecting DNA virus, the geminivirus tomato yellow leaf curl virus (TYLCV), physically associate with one another in an intricate network, as detected by a number of protein-protein interaction techniques. Importantly, our results indicate that intra-viral protein-protein interactions can modify the subcellular localization of the proteins involved. Using one particular pairwise interaction, that between the virus-encoded C2 and CP proteins, as proof-of-concept, we demonstrate that the combination of viral proteins leads to novel transcriptional effects on the host cell. Taken together, our results underscore the importance of studying viral protein function in the context of the infection. We propose a model in which viral proteins might have evolved to extensively interact with other elements within the viral proteome, enlarging the potential functional landscape available to the pathogen.

## Author summary

Viruses are obligate intracellular parasites that depend on the molecular machinery of their host cell to complete their life cycle. For this purpose, viruses co-opt host processes, modulating or redirecting them. Most viruses have small genomes, and hence limited

**Funding:** This work was supported by the Strategic Priority Research Program of the Chinese Academy of Sciences (CAS) (grant number XDB27040206), the Shanghai Center for Plant Stress Biology, CAS, and the Excellence Strategy of the German Federal and State Governments to RL-D. RL-D was the recipient of a National Foreign Talents project (grant number G20200113006). LW is the recipient of a Young Investigator Grant from the Natural Science Foundation of China (NSFC) (grant number 32100249). LM-P was the recipient of a Young Investigator Grant from NSFC (grant number 31850410467), a President's International Fellowship Initiative (PIFI) postdoctoral fellowship (2018PB058 and 2020PB0080) from CAS, and a Foreign Youth Talent Program project (grant number 20WZ2503900) from the Shanghai Science and Technology Commission. BGG was the recipient of a President's International Fellowship Initiative (PIFI) postdoctoral fellowship (2020PB0082), and a Foreign Youth Talent Program project (grant number 20WZ2504500) from the Shanghai Science and Technology Commission. The funders had no role in study design, data collection and analysis, decision to publish, or preparation of the manuscript.

**Competing interests:** The authors have declared that no competing interests exist.

coding capacity. During the viral invasion, virus-encoded proteins will be produced in large amounts and coexist in the infected cell, which enables physical or functional interactions among viral proteins, potentially expanding the virus-host functional interface by increasing the number of potential targets in the host cell and/or synergistically modulating the cellular environment. Examples of interactions between viral proteins have been recently documented for both animal and plant viruses; however, the hypothesis that viral proteins might have a combinatorial effect, which would lead to the acquisition of novel functions, lacks systematic experimental validation. Here, we use the geminivirus tomato yellow leaf curl virus (TYLCV), a plant-infecting virus with reduced proteome and causing devastating diseases in crops, to test the idea that combinatorial interactions between viral proteins exist and might underlie an expansion of the functional landscape of the viral proteome. Our results indicate that viral proteins prevalently interact with one another in the context of the infection, which can result in the acquisition of novel functions.

## Introduction

Viruses are intracellular parasites that need to subvert the host cell in order to enable their own replication and ensure viral spread. For this purpose, viruses co-opt the cell molecular machinery, modulating or redirecting its functions; as a result, infected cells undergo dramatic molecular changes, including heavy transcriptional reprogramming, concomitant to the proliferation of the virus.

Most viruses have small genomes, likely due to bottlenecks in encapsidation and/or within-host transport, which imposes limitations in coding capacity: viral proteins frequently exhibit small size, and their numbers per viral genome range from a few (<10) to a few dozen (Fig 1). Viral proteins have evolved to be multifunctional, and have been suggested to target hubs in the proteomes of their host cells [1–4], hence maximizing the impact of the viral-host protein-protein interactions; nevertheless, how a limited repertoire of small viral proteins can lead to the drastic cellular changes observed during the viral infection remains puzzling. Upon viral invasion, virus-encoded proteins are produced in large amounts in the infected cells, where they co-exist. Therefore, physical or functional interactions among viral proteins might have evolved as a potential mechanism to expand the virus-host functional interface, increasing the number of potential targets in the host cell and/or synergistically modulating the cellular environment. Interestingly, examples of interactions between viral proteins have been recently documented for both animal and plant viruses (e.g. [5–26]; see VirHostNet 2.0, http://virhostnet.prabi.fr/ [27]); some of these interactions are proposed to contribute to viral genome replication and virion assembly. However, the hypothesis that the combination of individual virus-encoded proteins might result in the acquisition of novel functions still lacks experimental support, and therefore the general biological relevance of these protein-protein interactions remains unclear.

Here, we use the plant DNA virus tomato yellow leaf curl virus (TYLCV; Fam. *Geminiviridae*) to test the idea that combinatorial interactions among viral proteins exist and may underlie an expansion of the functional landscape of the viral proteome. TYLCV is traditionally believed to encode six proteins (C1/Rep, C2, C3, C4, V2, and CP), although additional open reading frames (ORFs) have been recently described in its genome [28]. Local infection by TYLCV in the experimental host *Nicotiana benthamiana* results in heavy transcriptional reprogramming, with >11,000 differentially expressed genes (DEGs) detected at 6 days post-

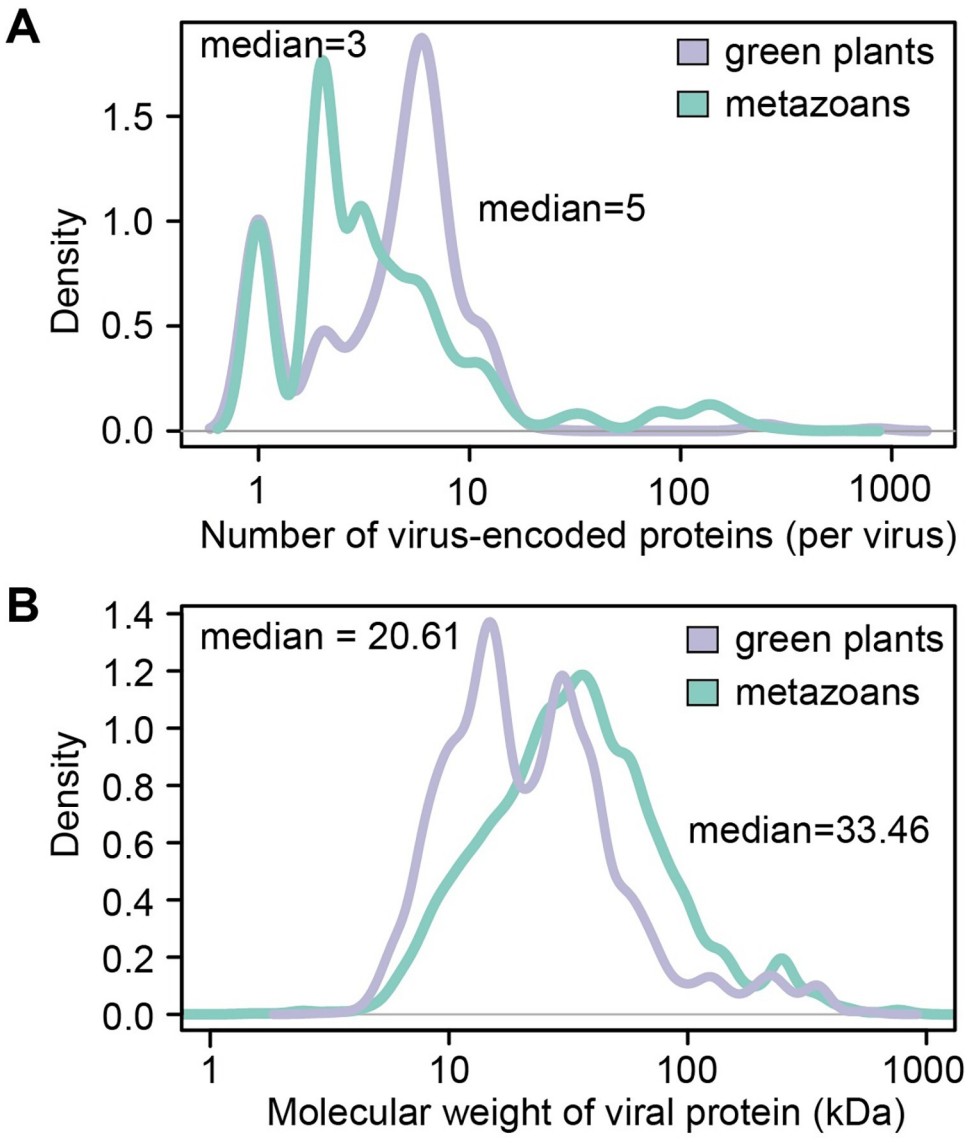

**Fig 1. Average number of virus-encoded proteins and their molecular weight. (A)** Average numbers of virus-encoded proteins (per virus) in animal and plant viruses. **(B)** Molecular weight of viral proteins from animal and plant viruses. Sequences in (A and B) were downloaded from NCBI Virus, from complete RefSeq genome sequences of viruses infecting *Viridiplantae* (green plants, taxid: 33090) or *Metazoa* (metazoans, taxid: 33208).

inoculation (dpi) [29]. Although a limited number of viral protein-protein interactions have been described for this virus to date [30–35], the intra-viral interactome has not been systematically explored, and the functional impact of these interactions remains elusive. Our results show that, strikingly, viral proteins form prevalent pairwise interactions in the context of the viral infection, displaying a high degree of intra-viral connectivity, as demonstrated by an array of protein-protein interaction techniques. As proof-of-concept for the idea that intra-viral protein-protein interactions can expand the functional repertoire of individual viral proteins, we focus on the pair formed by C2 and CP, since the presence of the latter is required and sufficient to shift the subcellular localization of the former from the nucleoplasm to the nucleolus, where the interaction occurs. Our data indicate that the combination of C2 and CP

results in drastic transcriptional reprogramming in the host plant, which goes beyond the sum of the effects of each of the individual proteins, hence supporting the idea that combinatorial interactions between viral proteins, physical and/or functional, can expand the functional repertoire of the viral proteome. The results obtained here might have important implications for the study not only of plant-geminivirus interactions, but of viral infections in general.

## Results and discussion

### Viral proteins form complexes in the host cell

In order to test whether virus (TYLCV)-encoded proteins associate with one another, we employed a number of protein-protein interaction methods, namely yeast two-hybrid (Y2H), and *in planta* co-immunoprecipitation (co-IP), bimolecular fluorescence complementation (BiFC), and split-luciferase assays. Several viral protein-protein interactions were identified in yeast (Figs 2A and S1) (Rep-Rep, Rep-C3, C2-C3, C2-C4, C3-C3, C3-C4, C3-CP, C3-V2, C4-C3, V2-V2); of note, the C2-C2 self-interaction could not be evaluated, since full-length C2 fused to the GAL4 binding domain displays auto-activation (Fig 2A), as previously described for other geminiviral C2 proteins [36–37]. Next, pairwise interactions between viral proteins were tested by co-IP assays following transient expression of C-terminally tagged versions of the viral proteins in *N. benthamiana*. The number of associations between viral proteins found in co-IP, which detects both direct and indirect interactions, was higher (Figs 2B, S2 and S3). The viral infection reshapes the cell environment where the viral proteins coexist: the presence/absence of host proteins, the activation of post-translational modification pathways, or the presence of additional viral proteins might influence the outcome of the tested interactions. Therefore, co-IP assays were performed both in the presence and absence of the virus. Most reproducible interactions detected in the absence of the virus were maintained in the context of the infection (Rep-Rep, C2-C2, C4-C2, C4-C3, C4-C4, C4-V2, CP-C2, V2-C2, V2-V2), and additional interactions were detected in an infection-dependent manner (Rep-C2, Rep-C3, Rep-C4, Rep-CP, Rep-V2, CP-Rep, C4-CP, CP-CP, CP-V2, V2-C3). Viral protein-protein interactions were further tested *in planta* by BiFC and split-luciferase assays (Fig 2C and 2D). In both assays, the viral proteins were fused to one half of the protein to be reconstituted upon a positive interaction (nYFP and cYFP for YFP, or nLuc and cLuc for luciferase, respectively) and transiently expressed in *N. benthamiana* leaves; for the BiFC experiments, n-YFP and c-YFP were fused to the C-terminus of the viral proteins, while for split-luciferase experiments nLuc was fused to the C-terminus and cLuc to the N-terminus of the viral protein. Interestingly, BiFC indicates that most of the detected interactions occur in the nucleus, with different distribution patterns, including localization in the nucleoplasm (e.g. Rep-C2), nucleolus (e.g. C3-C3), or nuclear speckles (e.g. Rep-C3) (Figs 2C and S4; additional patterns of interactions observed by BiFC can be found in S5 Fig). This nuclear prevalence of viral protein-protein interactions correlates with the nucleus hosting most of the viral cycle, including replication, transcription of viral genes, and encapsidation [38]. One notable exception is the interaction between C4 and V2, which takes place in intracellular punctate structures outside of the nucleus; this localization may be linked to the proposed role of these proteins in viral movement [39]. All viral proteins were shown to interact with one another (including self-interactions) at least in one direction by BiFC (Fig 2C). Similarly, all viral proteins displayed a positive interaction with each of the viral proteins by split-luciferase assays, with three exceptions (Rep-CP, C4-CP, and V2-CP) (Fig 2D). The identification of positive interactions in these reconstitution assays that were not detected as such by co-IP could be explained by the potential weak or transient nature of these associations, which could be overcome by artificial stabilization provided by the complementation of the split reporter.

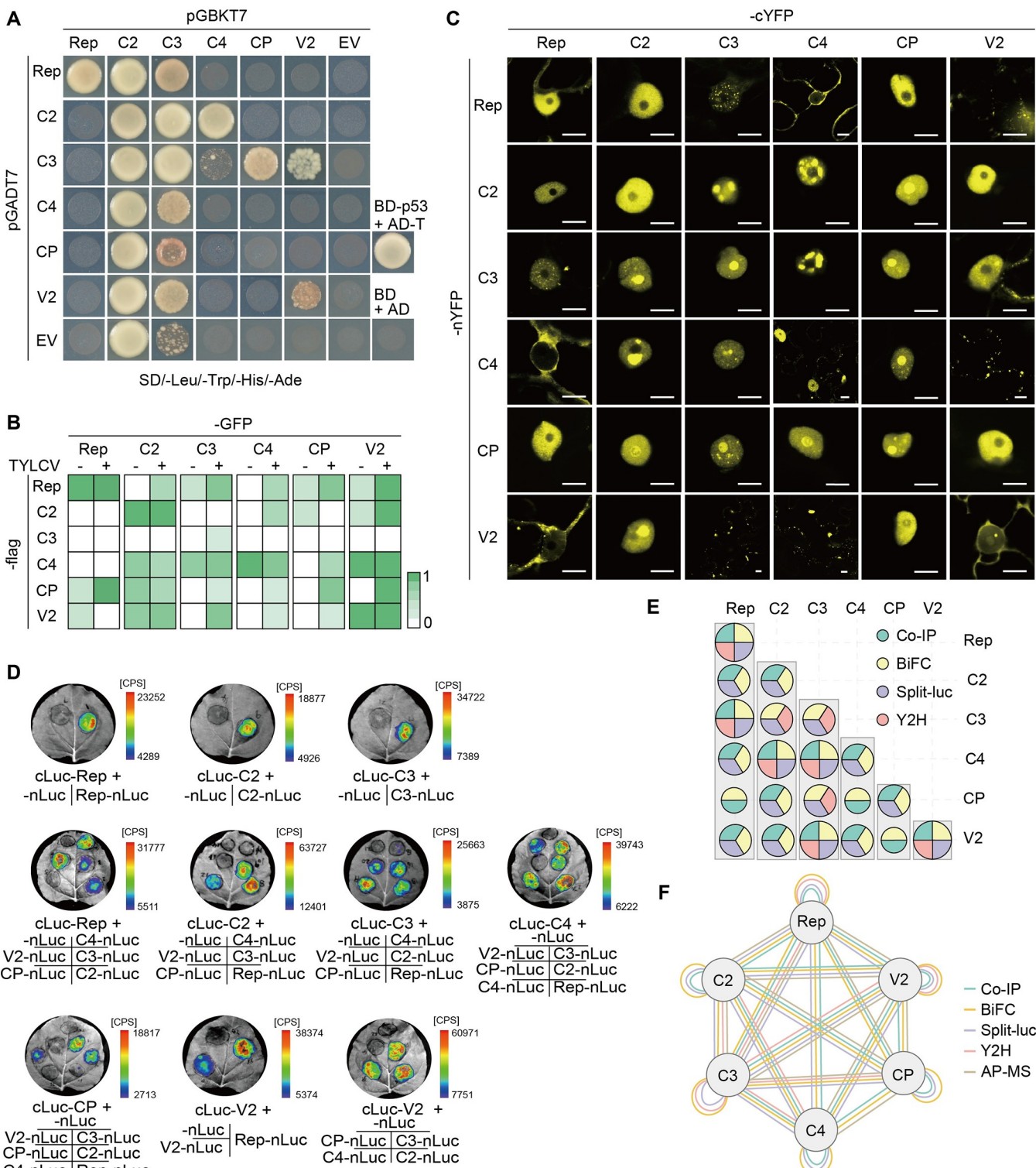

**Fig 2. The proteins encoded by the plant DNA virus tomato yellow leaf curl virus (TYLCV) associate with one another in the plant cell. (A)** Viral protein-protein interactions detected in yeast two-hybrid. The minimal synthetic defined (SD) medium without leucine (Leu), tryptophan (Trp), histidine (His), and adenine (Ade) was used to select positive interactions; SD without Leu and Trp was used to select co-transformants (S1 Fig). The interaction between the SV40 large T antigen (T) and the murine tumor suppressor p53 is a positive control. AD: activation domain; BD: binding domain. This experiment was repeated twice with similar results. **(B)** Summary of viral protein-protein interactions detected by co-immunoprecipitation (co-IP) in the absence (-) or presence (+) of TYLCV. These experiments were repeated at least three times; the colour scale represents the percentage of positive interaction results among all replicates,

with 1 = 100%. The original co-IP blots are shown in S2 Fig (in the absence of TYLCV) and S3 Fig (in the presence of TYLCV). An interaction between two viral proteins was considered as positive if at least two replicates showed positive interactions either in the absence or presence of TYLCV. **(C)** Viral protein-protein interactions detected by bimolecular fluorescence complementation (BiFC) in *N. benthamiana* leaves. nYFP: N-terminal half of the YFP; cYFP: C-terminal half of the YFP. Images were taken at 2 days post-infiltration (dpi). Scale bar = 10 μm. This experiment was repeated at least four times; combination with Hoechst staining and negative controls can be found in S4 Fig. Additional images are shown in S5 Fig. **(D)** Viral protein-protein interactions detected by split-luciferase assay in *N. benthamiana* leaves. nLuc: N-terminal part of the luciferase protein; cLuc: C-terminal part of the luciferase protein. Images were taken at 2 dpi. The colour scale represents the intensity of the interaction in counts per second (CPS). This experiment was repeated three times with similar results. **(E)** Summary of the intra-viral protein-protein interactions identified in **(A-D)**. Different colours represent different methods, as indicated; circle size indicates the number of the methods in which a positive interaction was detected. **(F)** Network of intra-viral protein-protein interactions. The colored lines indicate the positive interactions detected by Y2H, Co-IP, BiFC, split-luciferase assay, or AP-MS. See also S1–S5 Figs; S1 Table.

A summary of all detected pairwise interactions between viral proteins is shown in Fig 2E and 2F; all viral proteins were found to interact with one another, including self-interactions, by at least two independent methods. Importantly, some of these interactions could also be detected in unbiased affinity purification followed by mass spectrometry (AP-MS) experiments with C-terminal GFP-tagged versions of the viral proteins expressed in infected *N. benthamiana* cells [40] when these datasets were re-analyzed to search for the viral proteins, indicating that viral proteins physically associate with one another in the context of the infection in their native state (S1 Table).

As shown in Fig 2E and 2F, only certain pairwise viral protein-protein interactions could be detected by all used approaches (Rep-Rep, Rep-C3, C2-C4, C3-C4, C3-V2, and V2-V2). The differences in the detected outcomes when using different techniques might be due to specific requirements of each of the assays (e.g. strength or stability of the interaction necessary for this to be detected), or to the effect of the tags used (nature and position) on the interaction.

## The viral coat protein is required and sufficient to modify the subcellular localization of the virus-encoded C2 protein

Although the proteins encoded by TYLCV display specific localizations in the plant cell, all of them, with the exception of C4 (at the plasma membrane and weakly in chloroplasts), can be clear and consistently found in the nucleus (nucleoplasm and/or subnuclear compartments) in basal conditions when expressed alone fused to GFP (Rep: nucleoplasm; C2: nucleoplasm; C3: nucleoplasm, nucleolus, and nuclear speckles; CP: nucleolus and weakly in the nucleoplasm; V2: Cajal body and weakly in the nucleoplasm–in addition to endoplasmic reticulum) (Fig 3A). Interestingly, in the presence of the virus, several viral proteins fused to GFP, namely C2, C3, C4, and CP, experienced noticeable changes in their subcellular distribution (Fig 3A): C2-GFP, which is excluded from the nucleolus in the absence of the virus, accumulates in this compartment in infected cells; C3-GFP, on the contrary, is excluded from the nucleolus in the presence of the virus; C4-GFP is depleted from the plasma membrane and accumulates in chloroplasts; and CP-GFP re-localizes from the nucleolus to the nucleoplasm, where it accumulates in unidentified structures. These changes had been previously reported for C4-GFP and CP-GFP; while in the case of C4-GFP, Rep alone can trigger its re-localization from the plasma membrane to chloroplasts [41], no individual protein was sufficient to modify the subnuclear pattern of CP [31].

Using transient co-expression of C2-GFP and each viral protein fused to RFP at their C-terminus in *N. benthamiana* leaves, we could determine that CP is sufficient to enable a strong accumulation of C2 in the nucleolus, an effect that can also be triggered by an untagged version of CP (Figs 3B and 3C; S6). Of note, C2 and CP have been shown to interact in the nucleolus (Fig 2C). Curiously, only C2-GFP, but not GFP-C2, re-localizes to the nucleolus when in

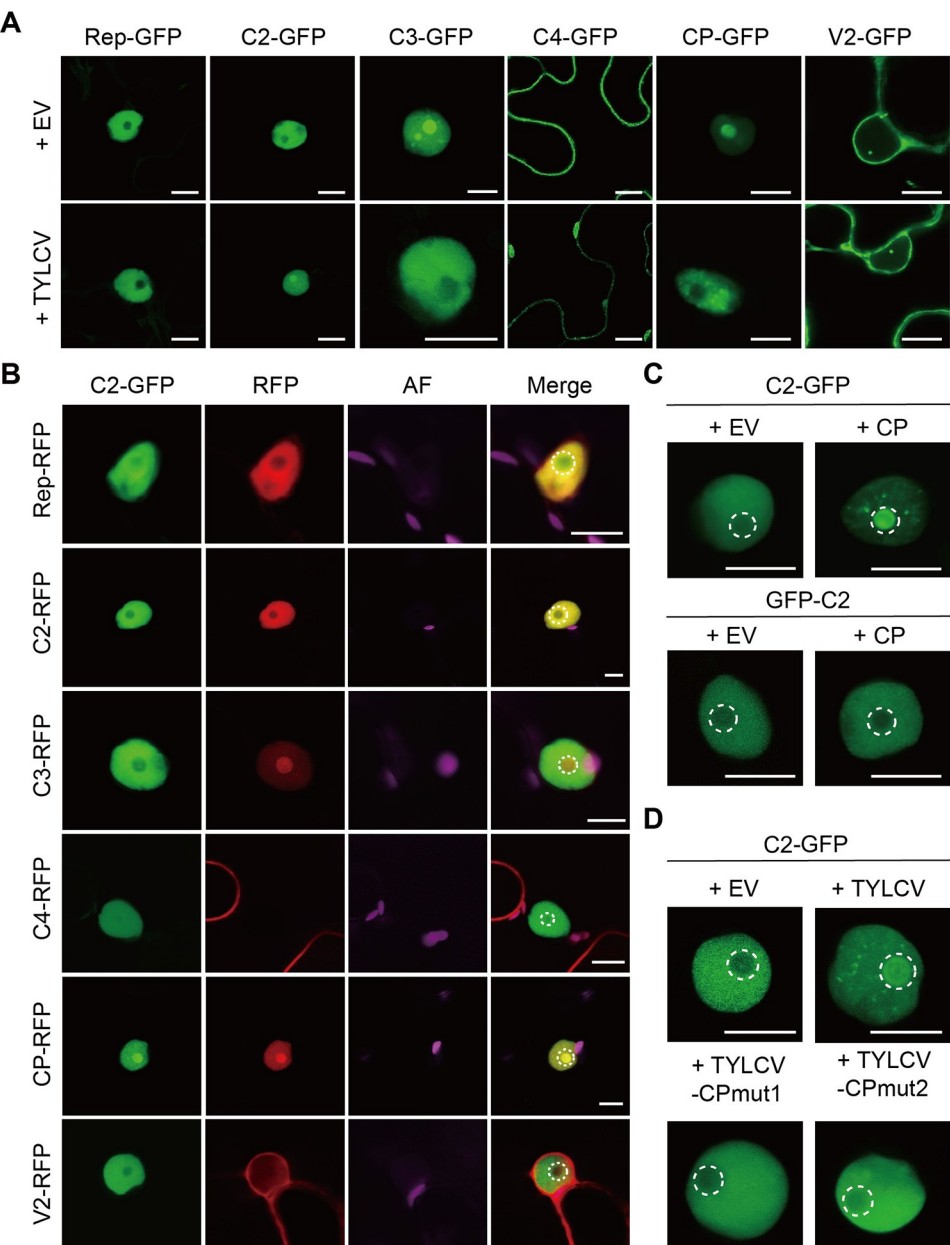

**Fig 3. CP is required and sufficient to change the subnuclear localization of C2. (A)** Subcellular localization of the TYLCV-encoded proteins fused to GFP at their C-terminus expressed alone (+EV; co-transformed with an empty vector control) or in the context of the viral infection (+TYLCV; co-transformed with a TYLCV infectious clone) in *N. benthamiana* leaves at 2 days post infiltration (dpi). Scale bar = 10 μm. EV: empty vector. **(B)** Subcellular localization of C2-GFP co-expressed with each of the viral proteins fused to RFP in *N. benthamiana* leaves at 2 dpi. Scale bar = 10 μm. AF: Autofluorescence. **(C)** Subcellular localization of C2-GFP or GFP-C2 when expressed alone (+EV) or co-expressed with CP (+CP) in *N. benthamiana* leaves at 2 dpi. The accumulation of the CP transcript is shown in S6A Fig. Scale bar = 10 μm. EV: empty vector. **(D)** Subcellular localization of C2-GFP when expressed alone (+EV) or in the context of the infection by the WT TYLCV virus (+TYLCV) or mutated versions unable to produce CP (+TYLCV-CPmut1; +TYLCV-CPmut2) in *N. benthamiana* leaves at 2 dpi. Scale bar = 10 μm. EV: empty vector. Viral accumulation is shown in S6B Fig. For details on TYLCV-CPmut1 and TYLCV-CPmut2, see Materials and Methods. In (**B**-**D**), the dashed circles mark the nucleolus. See also S6 Fig.

 

the presence of CP, likely due to a positional effect of the GFP tag (Fig 3C). Although full functionality of C2-GFP or GFP-C2 during the viral infection has not been demonstrated, C-terminal GFP fusions have been used in functional studies with other geminiviral C2 proteins (e.g. [42–45]). Local infection assays with two TYLCV mutant viruses unable to express CP, TYLCV-CPmut1 and TYLCV-CPmut2, in which early stop codons are introduced and alternative transcriptional initiation sites have been removed (for details, see Materials and Methods), demonstrate that the presence of CP is not only sufficient, but also required for the re-localization of C2-GFP into the nucleolus in infected cells (Fig 3D). These mutants accumulate to wild type-like levels in the transiently transformed leaves (S6B Fig).

Since, in the absence of the virus, C2-GFP appears evenly distributed in the nucleoplasm and is excluded from the nucleolus, but it gains strong nucleolar accumulation in the presence of the virus, we reasoned that C2 might perform additional functions in the context of the infection, and decided to use the C2-CP interaction as a proof-of-concept for the idea that viral proteins might have combinatorial functions.

## The C2/CP module specifically reshapes the host transcriptome

With the purpose of assessing if the functional landscape of C2 might be expanded when in the presence of CP, and considering that the C2 protein from geminiviruses has been previously described to impact host gene expression [46–51], we decided to investigate the transcriptional changes triggered by C2 in the presence or absence of CP as a readout for the activity of the former. To this aim, we transiently expressed C2, CP, or C2+CP in *N. benthamiana* leaves and determined the resulting changes in the plant transcriptome by RNA sequencing (RNA-seq). As shown in Fig 4A, C2 alone caused the differential expression of 211 genes (139 up-regulated, 71 down-regulated), while expression of CP did not significantly affect the plant transcriptional landscape; simultaneous expression of C2 and CP resulted in a moderate increase in the number of differentially expressed genes (DEGs) to 263 (72 up-regulated, 191 down-regulated) (Figs 4A, S7A and S7B; S2 Table; validation of the RNA-seq results by RT-qPCR is presented in S7C Fig.). Strikingly, however, the identity and behavior of DEGs was dramatically changed by the presence of CP (Fig 4B and 4C), indicating that C2 and CP have a synergistic effect on the host transcriptome. Functional enrichment analysis unveiled that addition of CP indeed shifted the functional gene ontology (GO) categories transcriptionally reprogrammed by C2, and that certain categories appear as statistically over-represented in the subset of down-regulated genes only when both viral proteins are simultaneously expressed (Fig 4D and 4E; S3 Table).

To investigate the relevance of the re-localization of C2 to the nucleolus (Fig 3B and 3C) for this effect, we selected DEGs specifically affected by the co-expression of C2 and CP, and tested the ability of C2-GFP (which re-localizes to the nucleolus in the presence of CP) or GFP-C2 (which does not re-localize to the nucleolus in the presence of CP) to affect their transcript accumulation when transiently co-expressed with CP in *N. benthamiana* leaves, as measured by RT-qPCR. As shown in Fig 5, only C2+CP and C2-GFP+CP, but not GFP-C2+CP, affect the expression of the selected genes compared with C2, C2-GFP, or GFP-C2, respectively. This result suggests that the modification in subnuclear localization of C2 mediated by CP is likely required for the impact of the combination of these proteins on host gene expression.

Next, we investigated the contribution of C2 and CP to the virus-induced transcriptional reprogramming in the context of the viral infection. We reasoned that, if C2 and CP together affect the transcriptional landscape of the host in a different manner than C2 or CP alone, then the transcriptional changes triggered by mutated versions of the virus unable to produce either C2 or CP should present overlapping differences compared to the changes triggered by the

 

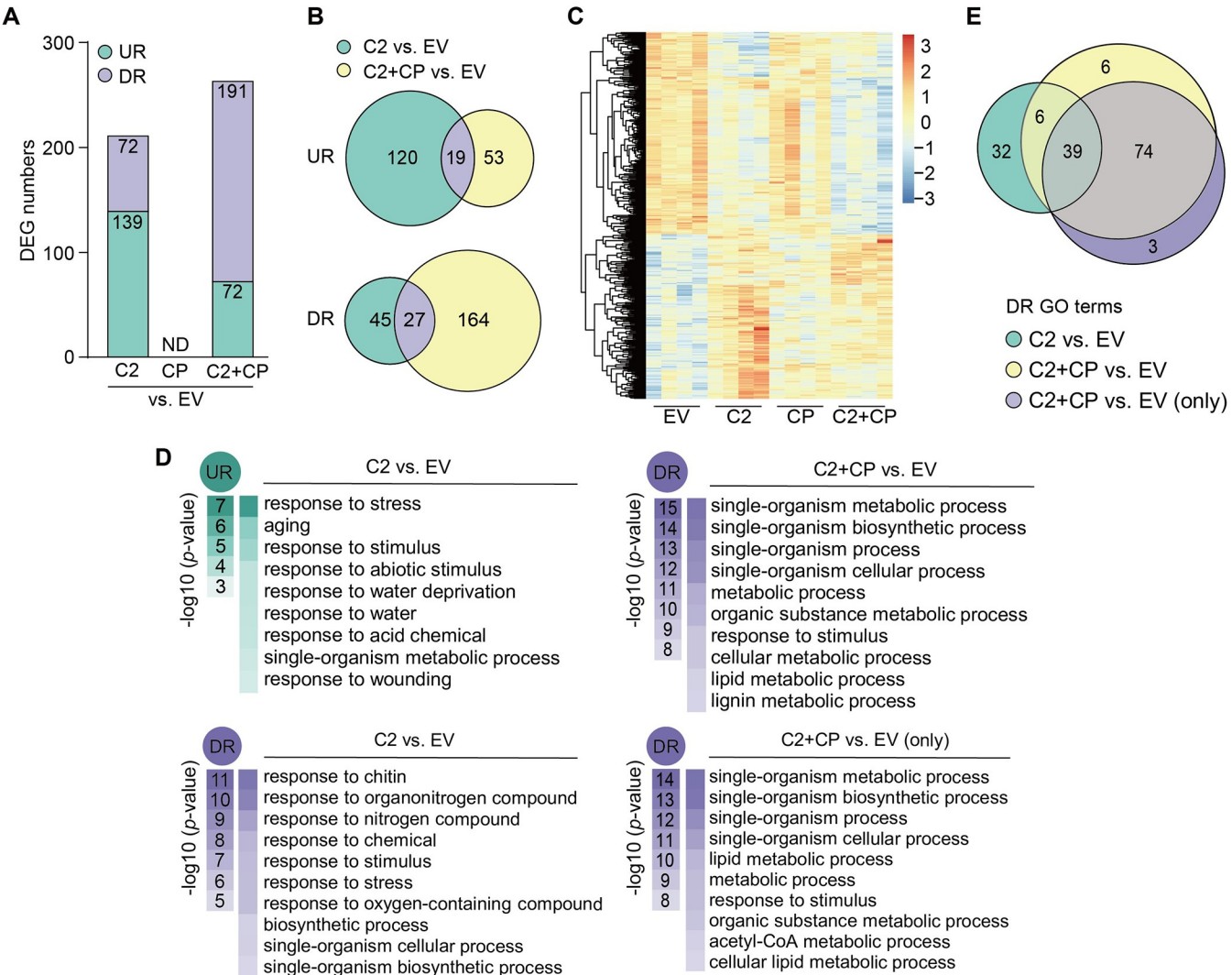

**Fig 4. C2 and CP functionally interact *in planta* and modify the transcriptome of *N. benthamiana* in an interdependent manner. (A)** Number of differentially expressed genes (DEGs) upon expression of C2, CP, or C2+CP in *N. benthamiana* leaves. UR: up-regulated; DR: down-regulated; ND: not detected; EV: empty vector. Full lists can be found in S2 Table. **(B)** Venn diagram of DEGs upon expression of C2 or C2+CP in *N. benthamiana*. UR: up-regulated; DR: down-regulated; EV: empty vector. **(C)** Heatmap with hierarchical clustering from samples in **(A)**. The colour scale indicates the Z-score. EV: empty vector. **(D)** Functional enrichment analysis of up-regulated (UR) or down-regulated (DR) genes in the indicated samples. Gene Ontology (GO) categories from the Biological Process ontology enriched with a *p*-value<0.01 (up to top 10) are shown; functional enrichment was performed using the orthologues in *Arabidopsis thaliana*. "C2+CP vs. EV (only)" denotes the subset of genes that are down-regulated in this sample only, and not in the samples expressing the viral proteins separately. The colour scale indicates the -log10 (*p*-value), showing the significance of GO term enrichment. EV: empty vector. For a full list, see S3 Table. **(E)** Venn diagram of the GO terms (Biological Process ontology) over-represented in the subsets of down-regulated genes (*p*-value<0.01) in the different samples. DR: down-regulated; EV: empty vector. For a full list, see S3 Table. See also S7 Fig; S2 and S3 Tables.

wild-type (WT) virus. Following this rationale, we compared the transcriptome of *N. benthamiana* leaves infected with the WT virus or mutated versions unable to produce C2 (TYLCV-C2mut) or CP (TYLCV-CPmut1), with respect to the empty vector (EV) control (Fig 6A) or to the WT virus (Fig 6B). As expected, both point mutants were unable to establish a full systemic infection, indicating that the corresponding viral proteins are most likely not produced from the mutated genes (S8A and S8B Fig). Of note, although the CP null mutant (TYLCV-CPmut1) accumulated to lower levels in these assays, no significant changes in the accumulation of viral transcripts were detected among these viral variants in local infection

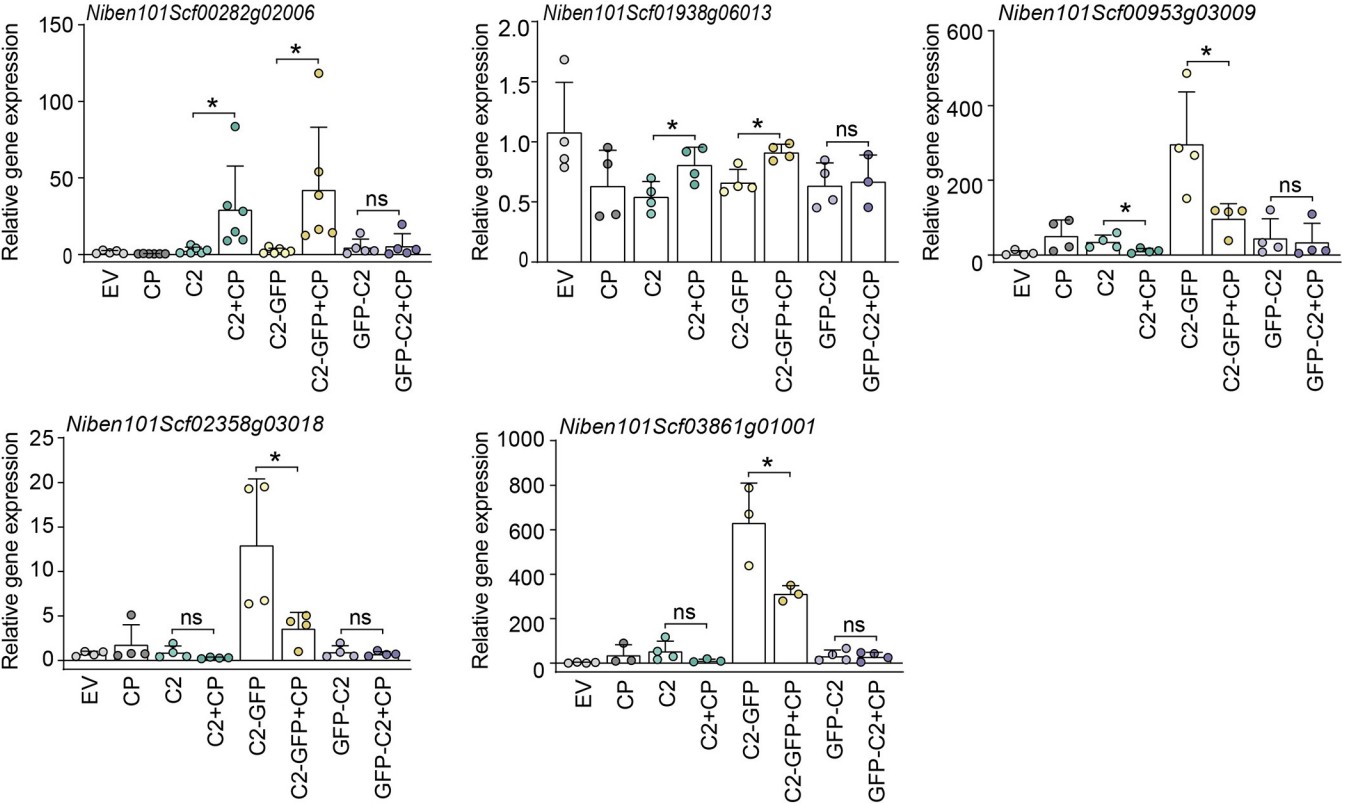

**Fig 5. Expression of selected DEGs upon transient expression of C2, C2-GFP, or GFP-C2 in the presence and absence of CP in *N. benthamiana* leaves.**
Gene expression was measured by RT-qPCR. The samples expressing CP or empty vector (EV) are used as control. Expression values are the mean of at least three biological replicates. Error bars represent SD. Asterisks indicate a statistically significant difference (*: $p<0.05$, **: $p<0.01$) according to a two-tailed comparison t-test. *NbACT2* was used as the normalizer.

assays (S6B and S8C–S8F Fig). Importantly, and despite the fact that expression of CP alone did not result in detectable transcriptional changes (Fig 4A), mutation of CP in the viral genome led to the differential expression of 3,256 genes when compared to the WT infection, supporting the notion that CP modulates host gene expression in combination, physical or functional, with other viral proteins; remarkably, 2,591 of these DEGs (79.5%) overlapped with those caused by the loss of C2 (Figs 6C, 6D and S8G; S2 Table; validation of the RNA-seq results is presented in S8H Fig), indicating that C2 and CP cooperatively mediate changes in host gene expression during the infection. Functional categories over-represented among the up-regulated genes in the presence of the WT virus appear as down-regulated in the subset of DEGs commonly triggered by the C2- and CP-deficient viruses compared to the WT version (Figs 6E and S9; S4 and S5 Tables), suggesting that the C2/CP module is responsible for the transcriptional changes of genes associated to these GO terms. A complete overview of the functional enrichment in the different subsets of DEGs can be found in S9 Fig and S5 Table.

Taken together, our results demonstrate that TYLCV proteins form an intricate network of interactions that potentially vastly increase the complexity of the virus-host interface, and that viral proteins can have additional effects on the host cell when in combination. Given that intra-viral protein-protein interactions have been reported for viruses belonging to independently evolved families and infecting hosts belonging to different kingdoms of life, we propose that this might be an evolutionary strategy of viruses to expand their functional repertoire

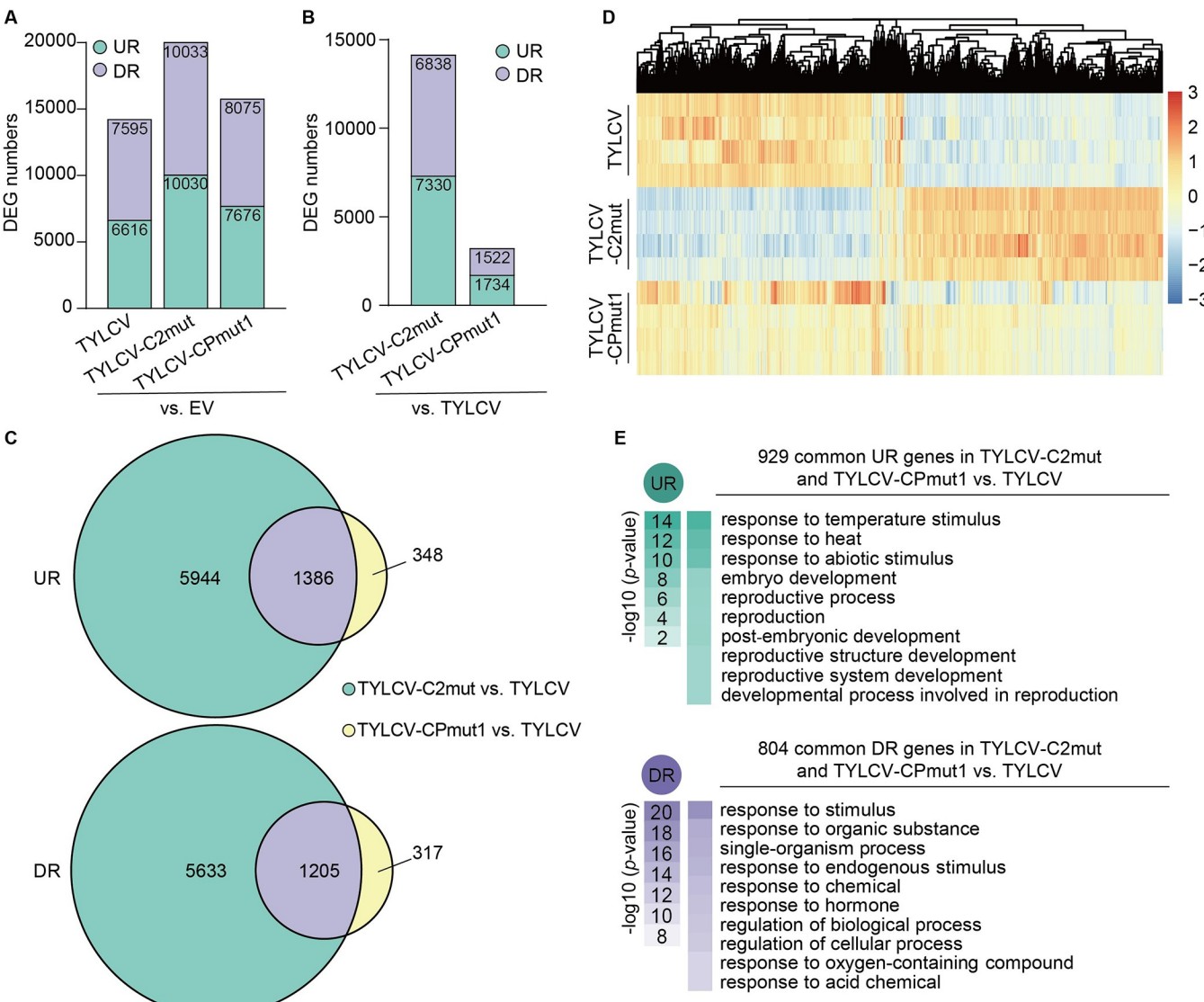

**Fig 6. C2 and CP functionally interact *in planta* in the context of the viral infection. (A, B)** Number of differentially expressed genes (DEGs) upon infection by TYLCV WT or C2-null or CP-null mutant variants (TYLCV-C2mut and TYLCV-CPmut1, respectively) in *N. benthamiana* leaves compared to the empty vector control **(A)**, or to TYLCV WT **(B)**. UR: up-regulated; DR: down-regulated; EV: empty vector. Full lists can be found in S2 Table. **(C)** Venn diagrams of DEGs upon infection by TYLCV C2-null and TYLCV CP-null mutants (TYLCV-C2mut and TYLCV-CPmut1, respectively) compared to TYLCV WT. UR: up-regulated; DR: down-regulated. **(D)** Heatmap with hierarchical clustering from **(A)**. The colour scale indicates the Z-score. **(E)** Functional enrichment analysis of the subsets of up-regulated (UR) or down-regulated (DR) genes in the indicated samples. Gene Ontology (GO) categories from the Biological Process ontology enriched with a *p*-value<0.01 (up to top 10) are shown; functional enrichment was performed using the orthologues in *A. thaliana*. The colour scale indicates the -log10 (*p*-value), showing the significance of GO term enrichment. For a full list, see S4 Table. See also S8 and S9 Figs; S2 and S4 Tables.

while maintaining small genomes, which would call for a reconsideration of our approaches to the study of viral protein function and virus-host interactions.

# Materials and methods

## Plant material

*Nicotiana benthamiana* plants were grown in a controlled growth chamber in long-day conditions (16 h light/8 h dark) at 25°C.

## Bacterial strains and growth conditions

*Escherichia coli* strain DH5α was used for general cloning and subcloning procedures; strain DB3.1 was used to amplify Gateway-compatible empty vectors.

For *in planta* expression, *Agrobacterium tumefaciens* strain GV3101 harbouring the corresponding binary vectors were liquid-cultured in LB medium (1% tryptone, 0.5% yeast extract, and 1% NaCl) with the appropriate antibiotics at 28˚C overnight.

## Plasmids and cloning

Open reading frames (ORFs, corresponding to Rep, C2, C3, C4, V2, and CP) from TYLCV (GenBank accession number AJ489258) were cloned in pENTR/D-TOPO (Thermo Scientific) with or without stop codon, to enable N- or C-terminal protein fusions, respectively [40]. The binary constructs to express viral proteins without tag, tagged with C-ter GFP, N-ter GFP, C-ter FLAG, or C-ter RFP, were generated by Gateway-cloning (LR reaction, Thermo Scientific) the TYLCV ORFs from pENTR/D-TOPO into pGWB2 [52], pGWB5 [52], pGWB6 [52], pGWB511 [53], and pGWB554 [53], respectively, with the exception of the construct to express C4-RFP, which was generated by Gateway-cloning the C4 ORF into pB7RWG2.0 [54]. For biomolecular fluorescence complementation assays (BiFC), the TYLCV ORFs were Gateway-cloned into pGTQL1211YN and pGTQL1221YC [55]. For yeast two-hybrid assays (Y2H), pGBKT7 and pGADT7 (Clontech) were digested with *EcoR*I and *Pst*I or *EcoR*I and *BamH*I, respectively, and the PCR-amplified Rep, C2, C3, C4, V2, and CP ORFs were in-fused to the C-terminus of the GAL4 DNA-binding domain (in pGBKT7) and the C-terminus of GAL4 activation domain (in pGADT7) with ClonExpress II One Step Cloning Kit (Vazyme). The binary constructs for split-luciferase complementation imaging assay were generated by Gateway cloning the TYLCV ORFs into pGWB-nLuc and pGWB-cLuc [56–57].

The TYLCV infectious clone has been previously described [58]. Using the wild-type (WT) infectious clone as template, the TYLCV C2 null mutant (TYLCV-C2mut), carrying a C-to-G mutation in the 14th nucleotide of the C2 ORF, was generated, converting the fifth codon (encoding a serine) to a stop codon, with the QuickChange Lightning Site-Directed Mutagenesis Kit (Agilent Technologies, Cat #210518). Similarly, the TYLCV CP null mutant 1 (TYLCV-CPmut1), carrying a C-to-A mutation in the fourth nucleotide of the CP ORF, was generated, converting the second codon (encoding a serine) to a stop codon. The TYLCV-CP-mut2 infectious clone, containing two premature stop codons in positions 2 and 15 and in which the nine potential alternative starting sites (ATG) have been removed, was synthesized. In both cases, the mutations in the CP ORF do not affect the overlapping V2 ORF.

All primers and plasmids used for cloning are summarized in S6 and S7 Tables, respectively.

## *Agrobacterium*-mediated transient gene expression in *N. benthamiana*

Transient expression assays were performed as previously described [40] with minor modifications. In brief, all binary plasmids were transformed into *A. tumefaciens* strain GV3101; *A. tumefaciens* clones carrying the constructs of interest were liquid-cultured in LB with the appropriate antibiotics at 28˚C overnight. Bacterial cultures were collected by centrifugation at 4,000 x g for 10 min and resuspended in the infiltration buffer (10 mM $MgCl_2$, 10 mM MES pH 5.6, 150 μM acetosyringone) to an $OD_{600}$ = 0.2–0.5. Next, bacterial suspensions were incubated at room temperature in the dark for 2–4 hours before infiltration into the abaxial side of 4-week-old *N. benthamiana* leaves with a 1 mL needleless syringe. For experiments that required co-infiltration, the *A. tumefaciens* suspensions carrying different constructs were mixed at 1:1 ratio before infiltration.

### Protein extraction and immunoprecipitation assays

Fully expanded young leaves of 4-week-old *N. benthamiana* plants were co-infiltrated with *A. tumefaciens* carrying constructs to express Rep-, C2-, C3-, C4-, CP-, and V2-FLAG, with Rep-, C2-, C3-, C4-, CP-, or V2-GFP. To analyze these protein-protein interactions in the context of the viral infection, *A. tumefaciens* carrying the infectious TYLCV clone were co-infiltrated in the respective experiments. Two days after infiltration, 0.7–1 g of infiltrated *N. benthamiana* leaves were harvested. Protein extraction, co-immunoprecipitation (co-IP), and western blot were performed as previously described [59]. For western blot, the following primary and secondary antibodies were used at the indicated dilutions: mouse anti-green fluorescent protein (GFP) (M0802-3a, Abiocode, Agoura Hills, CA, USA) (1:10,000), rabbit polyclonal anti-FLAG epitope (FLAG) (F7425, Sigma, St. Louis, MO, USA) (1:10,000), goat polyclonal anti-mouse coupled to horseradish peroxidase (A2554, Sigma, St. Louis, MO, USA) (1:15,000), and goat polyclonal anti-rabbit coupled to horseradish peroxidase (A0545, Sigma, St. Louis, MO, USA) (1:15,000).

### Bimolecular Fluorescence Complementation (BiFC)

Fully expanded young leaves of 4-week-old *N. benthamiana* plants were co-infiltrated with *A. tumefaciens* clones carrying the appropriate BiFC plasmids using a 1 mL needleless syringe and imaged two days post-infiltration with a Leica TCS SMD confocal microscope (Leica Microsystems) using the preset settings for YFP (Ex: 514 nm, Em: 525–575 nm). For nuclei staining, leaves were infiltrated with 5 μg/mL Hoechst 33258 (Sigma) solution and incubated in the dark for 30–60 minutes before observation by using the corresponding preset settings (Ex: 355 nm, Em: 430–480 nm).

### Yeast two-hybrid

pGBKT7- and pGADT7-based constructs were co-transformed into the Y2HGold yeast strain (Clontech) using Yeastmaker Yeast Transformation System 2 (Clontech) according to the manufacturer's instructions. The co-transformants were selected on minimal synthetic defined (SD) media without leucine and tryptophan; interactions were tested on SD media without leucine, tryptophan, histidine, and adenine. pGADT7-T (expressing the SV40 large T-antigen) and pGBKT7-p53 (expressing murine p53) constructs were used as positive control; empty vectors were used as negative control.

### Split-luciferase complementation imaging assay

*A. tumefaciens* strains carrying the appropriate plasmids were agroinfiltrated into 4-week-old *N. benthamiana* plants using a 1 mL needleless syringe. Two days post-infiltration, the same leaves were infiltrated with 1 mM D-luciferin solution and kept in the dark for 5 min before imaging. The luminescence images were captured using a CCD camera (NightShade LB 985, Berthold). For a full protocol, see [57].

### Visualization of protein subcellular localization

For subcellular localization, plant tissues expressing GFP- or RFP-fused proteins were imaged with a Leica TCS SP8 confocal microscope (Leica Microsystems) using the preset settings for GFP (Ex: 488 nm, Em: 500–550 nm) or RFP (Ex: 561 nm, Em: 580–630 nm).

   Confocal imaging for co-localization of C2-GFP and TYLCV proteins fused to RFP was performed on a Leica TCS SP8 point scanning confocal microscope using the pre-set

sequential scan settings for GFP (Ex:488 nm, Em:500–550 nm) and RFP (Ex:561 nm, Em:600–650 nm).

## TYLCV infection

For TYLCV local infection assays, fully expanded young leaves of 4-week-old *N. benthamiana* plants were infiltrated with *A. tumefaciens* carrying the TYLCV infectious clone (WT or mutants). Samples were collected at 2.5 days post-inoculation (dpi) to detect viral accumulation.

For TYLCV systemic infection assays, *A. tumefaciens* clones carrying the TYLCV infectious clones (WT or mutants) were syringe-inoculated in the stem of 2-week-old *N. benthamiana* plants. Leaf discs from the three youngest apical leaves were harvested at 21 dpi to detect viral accumulation.

## Determination of viral accumulation by quantitative PCR (qPCR)

To determine viral accumulation, total DNA was extracted from *N. benthamiana* leaves using the CTAB method [60]. The DNA from local infection assays was treated with *Dpn*I at 37˚C for 1 hour prior to further analysis. Quantitative PCR (qPCR) was performed with primers to amplify Rep [31]. The qPCR reaction was performed with Hieff qPCR SYBR Green Master Mix (Yeasen), with the following program: 3 min at 95˚C, and 40 cycles consisting of 15 s at 95˚C, 30 s at 60˚C. As internal reference for DNA detection, the *25S ribosomal DNA interspacer* (ITS) was used [61]. qPCR was performed in a BioRad CFX96 real-time system as described previously [31]. The primers used are described in S8 Table.

## Reverse transcription quantitative PCR (RT-qPCR)

RNA was extracted using the Plant RNA kit (OMEGA Bio-Tek); cDNA was prepared using the iScript gDNA Clear cDNA Synthesis Kit (Bio-Rad) according to the manufacturer's instructions. The qPCR reaction was performed with Hieff qPCR SYBR Green Master Mix (Yeasen), with the following program: 3 min at 95˚C, and 40 cycles consisting of 15 s at 95˚C, 30 s at 60˚C. *Elongation factor-1 alpha* (*NbEF1α*) [62] or *Actin2 (NbACT2)* [63] were used as reference genes, as indicated. The primers used are described in S8 Table.

## RNAseq and analysis

Transcriptome sequencing in *N. benthamiana* was performed as previously described [29]. Four biological replicates were used per sample. The paired-end reads were cleaned by Trimimomatic [64] (version 0.36). Clean read pairs were retained for further analysis after trimming the adapter sequence, removing low quality bases, and filtering short reads. The *N. benthamiana* draft genome sequence (v1.0.1) [65] was downloaded from the Sol Genomics Network (https://solgenomics.net/ftp/genomes/Nicotiana_benthamiana/assemblies/). Clean reads were mapped to the genome sequence by HISAT [66] (version 2.1.0) with default parameters. The number of reads that were mapped to each *N. benthamiana* gene was calculated with the htseq-count script in HTSeq [65]. Differentially expressed genes (DEGs) with at least 1.5 fold change in expression and a FDR < 0.05 between control and experiment samples were identified by using EdgeR [67].

The heatmap with hierarchical clustering was drawn by R package pheatmap. Venn diagrams were drawn by Venny (http://bioinformatics.psb.ugent.be/webtools/Venn/) and modified in Adobe Illustrator. The *Arabidopsis thaliana* homologous genes of the DEGs identified

in *N. benthamiana* were used for Gene Ontology (GO) term enrichment analysis in AgriGO v2.0 [68].

## Supporting information

**S1 Fig. Yeast two-hybrid co-transformation control.** The minimal synthetic defined (SD) medium without leucine (Leu) and tryptophan (Trp) was used to select co-transformants. (TIF)

**S2 Fig. Representative co-immunoprecipitation (co-IP) assays. (A-F)** Co-immunoprecipitation (co-IP) assays of Rep-, C2-, C3-, C4-, CP- and V2-FLAG with Rep- (**A**), C2- (**B**), C3- (**C**), C4- (**D**), CP- (**E**) or V2-GFP (**F**) following transient expression in *N. benthamiana* leaves. IB: immunoblotting, IP: immunoprecipitation, CBB: Coomassie brilliant blue. Molecular weight of Rep-, C2-, C3-, C4-, CP-, and V2-GFP is 65, 42, 43, 38, 57, and 40 kDa, respectively; molecular weight of Rep-, C2-, C3-, C4-, CP-, and V2-FLAG are 41, 15, 16, 11, 30, and 14 kDa, respectively. Asterisks indicate the expected band for each protein. **(G)** Summary table containing the results of all co-IP replicates performed in the absence of the virus. Column headings indicate the viral protein used as bait; row headings indicate prey proteins. (TIF)

**S3 Fig. Representative co-immunoprecipitation (co-IP) assays in the presence of the virus. (A-F)** Representative co-immunoprecipitation (co-IP) assays of Rep-, C2-, C3-, C4-, CP- and V2-FLAG with Rep- (**A**), C2- (**B**), C3- (**C**), C4- (**D**), CP- (**E**) or V2-GFP (**F**) following transient expression in *N. benthamiana* leaves in the presence of the virus. IB: immunoblotting, IP: immunoprecipitation, CBB: Coomassie brilliant blue. Molecular weight of Rep-, C2-, C3-, C4-, CP- and V2-GFP is 65, 42, 43, 38, 57, and 40 kDa, respectively; molecular weight of Rep-, C2-, C3-, C4-, CP- and V2-FLAG are 41, 15, 16, 11, 30, and 14 kDa, respectively. Asterisks indicate the expected band for each protein. **(G)** Summary table containing the results of all co-IP replicates performed in the presence of the virus. Column headings indicate the viral protein used as bait; row headings indicate prey proteins. (TIF)

**S4 Fig. Viral protein-protein interactions detected by bimolecular fluorescence complementation (BiFC) in *N. benthamiana* leaves combined with Hoechst staining (A) and negative controls (B).** nYFP: N-terminal half of the YFP; cYFP: C-terminal half of the YFP; BF: bright field. Images were taken at 2 days post-infiltration (dpi). Scale bar = 10 μm in (**A**) or 50 μm in (**B**). This experiment was repeated at least four times with similar results. (TIF)

**S5 Fig. Additional BiFC images.** nYFP: N-terminal half of the YFP; cYFP: C-terminal half of the YFP. Images were taken at 2 days post-infiltration (dpi). Scale bar = 10 μm. This experiment was repeated at least four times with similar results. (TIF)

**S6 Fig. Accumulation of the CP transcript and viral accumulation in the samples from Fig 3C and 3D, respectively. (A)** Accumulation of the CP transcript (from Fig 3C), measured by RT-qPCR. *NbEF1α* was used as the normalizer. Values represent the mean of three plants. Error bars represent SD. **(B)** Accumulation of viral DNA in samples from Fig 3D. ITS was used as the normalizer. Values represent the mean of three plants. Error bars represent SD. (TIF)

**S7 Fig. Validation of the expression of plant and viral genes in the samples used for RNA-seq (Fig 4). (A-B)** C2 and CP transcript accumulation, measured by RT-qPCR. Expression values are relative to *NbACT2*. Results are the mean of four biological replicates. Error bars represent SD. **(C)** Comparison of the accumulation of transcripts of selected DEGs in the RNA-seq data and as measured by RT-qPCR. Expression values are the mean of log2 FC, relative to EV, from four biological replicates. *NbACT2* was used as the normalizer. FC: fold change; EV: empty vector. **(D)** Expression of selected DEGs upon expression of C2, C2-GFP, GFP-C2 in the presence and absence of CP in *N. benthamiana* leaves. Samples expressing CP or empty vector (EV) are used as control. Expression values are the mean of at least three biological replicates. Error bars represent SD. Asterisks indicate a statistically significant difference (*: $p<0.05$) according to a two-tailed comparison t-test. *NbACT2* was used as the normalizer.
(TIF)

**S8 Fig. Validation of the expression of plant and viral genes in the samples used for RNA-seq (Fig 6). (A)** Symptoms in *N. benthamiana* plants inoculated with TYLCV WT or C2-/CP-null mutants (TYLCV-C2mut and TYLCV-CPmut1, respectively), or inoculated with empty vector (EV) as negative control. Pictures were taken at 21 days post-inoculation (dpi). Scale bar: 10 cm. **(B)** Viral DNA accumulation in systemic infections in *N. benthamiana* plants measured by qPCR. Values are the mean of three independent biological replicates. Error bars represent SD. Samples were taken at 21 dpi. ITS was used as the normalizer. **(C)** Viral DNA accumulation in *N. benthamiana* leaves infiltrated with TYLCV WT or C2-/CP-null mutants (TYLCV-C2mut and TYLCV-CPmut1, respectively), or transformed with empty vector (EV) as negative control. Samples were taken at 2.5 days post-inoculation (dpi). ITS was used as the normalizer. Values represent the mean of six plants. Error bars represent SD. **(D-E)** C2 and CP transcript accumulation. Expression values are relative to *NbACT2*. Results are the mean of four biological replicates. Error bars represent SD. **(F)** Expression of TYLCV genes in the different samples as detected by RNA-seq. RPKM: reads per kilobase of transcript per million mapped reads. **(G)** Venn diagram of the subsets of up- and down-regulated genes in the samples infected with TYLCV WT or C2-/CP-null mutants (TYLCV-C2mut and TYLCV-CPmut1) compared to the empty vector control (EV). UR: up-regulated; DR: down-regulated. **(H)** Expression of selected DEGs. Expression values are the mean of log2 FC, relative to samples inoculated with the EV, from four biological replicates. *NbACT2* was used as the normalizer. FC: fold change; EV: empty vector.
(TIF)

**S9 Fig. Functional enrichment analysis of the subsets of up-regulated (UR) or down-regulated (DR) genes in Fig 6A.** Gene Ontology (GO) categories from the Biological Process ontology enriched with a *p*-value<0.01 (up to top 10) are shown; functional enrichment analysis was performed using the orthologues in *A. thaliana*. The colour scale indicates the -log10 (*p*-value), showing the significance of GO term enrichment. For a full list, see S5 Table.
(TIF)

**S1 Table. AP-MS/MS data, related to Fig 2.**
(XLSX)

**S2 Table. DEGs in RNA-seq experiments, related to Fig 4 and 6.**
(XLSX)

**S3 Table. Functional enrichment analysis, related to Fig 4.**
(XLSX)

**S4 Table. Functional enrichment analysis, related to Fig 6.**
(XLSX)

**S5 Table. Functional enrichment analysis, related to S9 Fig.**
(XLSX)

**S6 Table. Primers used for cloning in this work.**
(XLSX)

**S7 Table. Plasmids used in this work.**
(XLSX)

**S8 Table. Primers used for qPCR and RT-qPCR in this work.**
(XLSX)

## Acknowledgments

The authors thank past and present members of the Lozano-Duran lab for fruitful discussions; Xinyu Jian, Aurora Luque, the PSC Core Cell Biology Facility, and the PSC Core Genomics Facility for technical assistance; and Alberto P Macho for critical reading of this manuscript.

## Author Contributions

**Conceptualization:** Rosa Lozano-Durán.

**Funding acquisition:** Rosa Lozano-Durán.

**Investigation:** Liping Wang, Huang Tan, Laura Medina-Puche, Mengshi Wu, Borja Garnelo Gomez, Man Gao, Chaonan Shi, Tamara Jimenez-Gongora, Pengfei Fan, Xue Ding, Dan Zhang, Yi Ding, Tábata Rosas-Díaz, Yujing Liu, Emmanuel Aguilar, Xing Fu.

**Supervision:** Rosa Lozano-Durán.

**Writing – original draft:** Rosa Lozano-Durán.

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
