## [Decision Letter · Decision Letter 0]

12 Jun 2022

Dear Dr Lozano-Durán,

Thank you very much for submitting your manuscript "Combinatorial interactions between viral proteins expand the potential functional landscape of the viral proteome" for consideration at PLOS Pathogens. As with all papers reviewed by the journal, your manuscript was reviewed by members of the editorial board and by several independent reviewers. In light of the reviews (below this email), we would like to invite the resubmission of a significantly-revised version that takes into account the reviewers' comments.

All reviewers agreed this paper has the potential to make a meaningful contribution to the field.  However, a complete re-write of the manuscript, with meaningful descriptions of how experiments were done and how results were interpreted, is required.  Shortcomings of methods (e.g., are the tagged proteins employed functional?) and inconsistencies in results must be noted and addressed.  In its present form, the manuscript appears to be written for a journal with page and figure limits, rendering it difficult to understand without consulting other papers and multiple supplemental figures.  All the experiments and data presented in the multi-component Figure 1, for example, are described in 16 lines of text.  Readers should be provided with sufficient information to evaluate results and conclusions.  Reviewer 3 provides some guidance on how to proceed.  The C2:CP interaction, which is the best-developed story, should perhaps be the major focus of the manuscript.

We cannot make any decision about publication until we have seen the revised manuscript and your response to the reviewers' comments. Your revised manuscript is also likely to be sent to reviewers for further evaluation.

Sincerely,

David M Bisaro, PhD

Associate Editor

PLOS Pathogens

Peter Nagy

Section Editor

PLOS Pathogens

Kasturi Haldar

Editor-in-Chief

PLOS Pathogens

orcid.org/0000-0001-5065-158X

Michael Malim

Editor-in-Chief

PLOS Pathogens

orcid.org/0000-0002-7699-2064

Reviewer's Responses to Questions

**Part I - Summary**

Reviewer #1: The manuscript entitled “combinatorial interactions between viral proteins expand the potential functional landscape of the viral proteome” by Wang and Tan, et al reported the systematic analysis of interactions among TYLCV proteins, the altered localization of certain viral proteins when combined with specific viral partners or in the presence of viral infection. Dramatically altered RNA transcriptomes were reported in combination of viral proteins with its partners (CP+C2) or viral infection compared to that in the presence of individual viral protein (C2 or CP). These results have promoted authors to conclude a combinatorial and synergistic effect on host expression reprogramming caused by the expression of two or several viral proteins or full infection.

Reviewer #2: The authors raise a meaningful study about the combinatorial interactions between viral proteins during infection. However, there are several questions needed further discussion.

Reviewer #3: The authors have used the geminivirus TYLCV, which has reduced proteome, to test the idea that combinatorial interactions between viral proteins exist and might underlie an expansion of the functions beyond the function of the single protein. Actually the general approach is to express viral proteins one by one to study their function. In this MS it is shown that viral proteins interact with one another in the context of the infection, which can result in the acquisition of novel functions. This approach to understand functions of viral proteins, although not completely new, is well appreciated.

This manuscript includes interesting results and accommodates plenty of experiments, but this reviewer found it difficult to read, mainly because the large amount of information inserted in each figure is not thoroughly described. Out of the 15 pages of text (Fig. legends, references, excluded) only 4 are dedicated to results and discussion.

**Part II – Major Issues: Key Experiments Required for Acceptance**

Reviewer #1: The findings are still in an early stage. Authors clearly showed that viral proteins relocalize in the presence of their partners or full infection. These relocalization correlated well with the significantly changed transcriptomes. However, these are nice observations without mechanistic understanding.

Reviewer #2: 1. It is reasonable to say that viral genomes have reduced coding capacity in the abstract. In fact, though the most viruses have limited genome, they are used to encode viral proteins at their maximum extent.

2. Fig 2C, is GFP-C2 or C2-GFP functional?

Reviewer #3: Title: a reader expects a lot from words like “functional landscape of viral proteome”. Actually, the best part of this MS is in the analysis of the pair C2 and CP. I suggest to tune-down the emphasis present in the title. And include the term geminivirus or TYLCV.

Actually, the core of this MS is in the chapter: “The C2/CP module specifically reshapes the host transcriptome”, while the other chapters could be revisited (and the figures reduced) in view of preparing the reader to this one.

This is a MS where so many different approaches have been used, with results not always in agreement with each other. This can, and should, be clearly stated as the readers should be able to understand the expriments,without the need of reading in detail several other previous papers.

Lines 125-127 and Fig. 2A: the case of V2 should be discussed, it is not IN the nucleus.

Line 129: “obvious” ? please explain.

Line 137 and Fig. 2B: the nucleolar localization is visible not only in C2/CP, but also, although partially, in C2/Rep and C2/C3: this is neither mentioned nor discussed.

Line 90 and Fig.1: “Our results show that, strikingly, viral proteins form prevalent pairwise interactions in the context of the viral infection, displaying a high degree of intraviral connectivity, as demonstrated by an array of protein-protein interaction techniques.” This is shown in Fig.1, which merits much deeper explanation and discussion, even in those cases where some data do not go in the direction expected. As an example, compare, in fig. 1, panel D (Summary of viral protein-protein interactions detected by co-immunoprecipitation (co-IP) in the absence or presence of TYLCV), with panel G (Summary of the intra-viral protein-protein interactions). In G which columns of D were taken? Why was it important to put in D both a column “-“ and one “+” TYLCV? Explain.

C2/C2 interaction is evident in Y2H (panel C), but is missing in panel G and in panel H: why?

**Part III – Minor Issues: Editorial and Data Presentation Modifications**

Reviewer #1: (No Response)

Reviewer #2: Fig 2A, virus infection can affect the subcellular localization of C3. It seems that C3 is excluded from nucleolus in the presence of virus infection. Please explain this why.

Title is too big and should more specific.

Reviewer #3: Colors in panel G of Fig.2 are not easily distinguishable, probably thicker lines, and/or a modification of colors, can improve visibility.

If there are pieces of this work that are from previous papers of the same authors, this should be clearly stated. One example is the IP-MS work, which apparently is from ref. 36.

Supplementary Table 1 merits a better legend.

PLOS authors have the option to publish the peer review history of their article (what does this mean?). If published, this will include your full peer review and any attached files.

Reviewer #1: No

Reviewer #2: No

Reviewer #3: No
---

## [Decision Letter · Decision Letter 1]

30 Sep 2022

Dear Dr Lozano-Durán,

We are pleased to inform you that your manuscript 'Combinatorial interactions between viral proteins expand the potential functional landscape of the tomato yellow leaf curl virus proteome' has been provisionally accepted for publication in PLOS Pathogens.

Best regards,

David M Bisaro, PhD

Associate Editor

PLOS Pathogens

Peter Nagy

Section Editor

PLOS Pathogens

Kasturi Haldar

Editor-in-Chief

PLOS Pathogens

orcid.org/0000-0001-5065-158X

Michael Malim

Editor-in-Chief

PLOS Pathogens

orcid.org/0000-0002-7699-2064

Reviewer Comments (if any, and for reference):

Reviewer's Responses to Questions

**Part I - Summary**

Reviewer #1: The manuscript “Combinatorial interactions between viral proteins enpand the potential functional landscape of the tomato yellow leaf curl virus proteome” by Wang et al reported the systematic analysis of interactions among TYLCV proteins, the altered localization of certain viral proteins when combined with specific viral partners or in the presence of viral infection. Dramatically altered RNA transcriptomes were reported in combination of viral proteins with its partners (CP+C2) or viral infection compared to that in the presence of C2 alone. These results have promoted authors to conclude a combinatorial and synergistic effect on host expression reprogramming caused by the expression of two or several viral proteins or full infection.

Reviewer #2: This revision has addressed my concerns and can be accpted.

Reviewer #3: issues raised in the previous version of the manuscript have been addressed properly

**Part II – Major Issues: Key Experiments Required for Acceptance**

Reviewer #1: none

Reviewer #2: (No Response)

Reviewer #3: //

**Part III – Minor Issues: Editorial and Data Presentation Modifications**

Reviewer #1: This revised version has nicely addressed all comments. I have some minor comments:

1. How did authors determine that the REP-C3 interaction occurred in nuclear speckles? Nucleoplasm and nucleolus are easy to recognize by readers but not nuclear speckles. Maybe a marker of nuclear speckles should be used?

2. Fig. 3B. In the Legends, it was stated “Subcellular localization of C2-GFP expressed alone or co-expressed with each of the viral proteins fused to RFPC2-GFP”. Which column did authors show images of C2-GFP alone in 3B? Or “C2-GFP alone” in the legends is a mistake?

3. Fig. 3A C4-GFP in the presence of TYLCV was described as in the chloroplasts. However, no chloroplasts were seen in the image. Or please point chloroplasts by arrows.

4. L179. Should state Fig 3A.

5. Remove “when” in L193.

Reviewer #2: (No Response)

Reviewer #3: (No Response)

PLOS authors have the option to publish the peer review history of their article (what does this mean?). If published, this will include your full peer review and any attached files.

Reviewer #1: No

Reviewer #2: No

Reviewer #3: **Yes: **Gian Paolo Accotto

---

## [Editor Report · Acceptance letter]

13 Oct 2022

Dear Dr Lozano-Durán,

We are delighted to inform you that your manuscript, " Combinatorial interactions between viral proteins expand the potential functional landscape of the tomato yellow leaf curl virus proteome ," has been formally accepted for publication in PLOS Pathogens.

Best regards,

Kasturi Haldar

Editor-in-Chief

PLOS Pathogens

orcid.org/0000-0001-5065-158X

Michael Malim

Editor-in-Chief

PLOS Pathogens

orcid.org/0000-0002-7699-2064